# Integrative Transcriptome, miRNAs, Degradome, and Phytohormone Analysis of *Brassica rapa* L. in Response to *Plasmodiophora brassicae*

**DOI:** 10.3390/ijms24032414

**Published:** 2023-01-26

**Authors:** Xiaochun Wei, Rujiao Liao, Xiaowei Zhang, Yanyan Zhao, Zhengqing Xie, Shuangjuan Yang, Henan Su, Zhiyong Wang, Luyue Zhang, Baoming Tian, Fang Wei, Yuxiang Yuan

**Affiliations:** 1Institute of Horticulture, Henan Academy of Agricultural Sciences, Graduate T&R Base of Zhengzhou University, Zhengzhou 450002, China; 2Henan International Joint Laboratory of Crop Gene Resources and Improvement, School of Agricultural Sciences, Zhengzhou University, Zhengzhou 450001, China

**Keywords:** *Plasmodiophora brassicae*, transcriptome, miRNAs, degradome, phytohormone, *Brassica rapa* L.

## Abstract

Clubroot is an infectious root disease caused by *Plasmodiophora brassicae* in *Brassica* crops, which can cause immeasurable losses. We analyzed integrative transcriptome, small RNAs, degradome, and phytohormone comprehensively to explore the infection mechanism of *P. brassicae.* In this study, root samples of *Brassica rapa* resistant line material BrT24 (R-line) and susceptible line material Y510-9 (S-line) were collected at four different time points for cytological, transcriptome, miRNA, and degradome analyses. We found the critical period of disease resistance and infection were at 0–3 DAI (days after inoculation) and 9–20 DAI, respectively. Based on our finding, we further analyzed the data of 9 DAI vs. 20 DAI of S-line and predicted the key genes *ARF8*, *NAC1*, *NAC4*, *TCP10*, *SPL14*, *REV,* and *AtHB*, which were related to clubroot disease development and regulating disease resistance mechanisms. These genes are mainly related to auxin, cytokinin, jasmonic acid, and ethylene cycles. We proposed a regulatory model of plant hormones under the mRNA–miRNA regulation in the critical period of *P. brassicae* infection by using the present data of the integrative transcriptome, small RNAs, degradome, and phytohormone with our previously published results. Our integrative analysis provided new insights into the regulation relationship of miRNAs and plant hormones during the process of disease infection with *P. brassicae.*

## 1. Introduction

The *Plasmodiophora brassicae* is an important biotrophic protist, which causes clubroot disease in cruciferous plants. This pathogen was first detected in Europe and is now found in all brassica-growing areas of the world; it seems to be getting worse everywhere [1]. The disease is harmful to Chinese cabbage, mustard, radish, cabbage, cauliflower, and other cultivated and wild species of Cruciferae and is difficult to prevent and control [2,3]. The resting spores of *P. brassicae* may maintain their infectiousness in the soil for up to 20 years [4,5]. In China, this disease was first reported in the 1950s in the southwest [6]. In recent years, the incidence of clubroot disease in China has been increasing year by year, especially in the southwest, where it has reached 65% of the planting area and seriously affected the production of cruciferous vegetables [7,8].

Plants are frequently infected by various pathogens, including fungi, bacteria, and viruses. The micro RNA (miRNA) is an important regulatory molecule of the plant, which acts as immune response to biological stress [9]. So far, it has been reported that some miRNAs are involved in plant defense against pathogens. When the host plant is attacked by bacteria and fungi, plants can trigger PAMP-triggered immunity (PTI) via pathogen-associated molecular pattern (PAMP) and effector-triggered immunity (ETI) to resist the infection of the pathogen by inducing the first and second layers of the plant immune system, respectively [10]. In *Arabidopsis*, miRNAs induce a short peptide of bacterial PAMP to resist infection of *Pseudomonas syringae* by down regulation of the auxin receptor *TIR1*, *AFB2,* and *AFB3*. miR160a participates and induces PTI by targeting *ARF*s and increasing callose deposition [11,12]. miRNA393 can be induced by *P. syringae* to increase the secretion of proteins related to the antibacterial course by inhibiting the expression of MEMB12, thereby promoting the ETI response [13]. It has been shown that many miRNAs are up-regulated and down-regulated under the stress of clubroot disease [14,15]. miR393*-TIR1/AFB3*, miR156-*SPLs*, miR160/miR167-*ARFs*, miR171-*SCLs*, and miR172-AP2/*TOEs*/*RAP2 -7* respond to clubroot stress by regulating different plant hormone levels. Some novel miRNAs, like novel_51/*CAC3* and_149/*LACS6,* are involved in fatty acid synthesis and peroxisome pathways. miR168/*Ago1* mediates host resistance to clubroot by participating in the plant PTI immune system [15,16]. These results suggest that miRNAs are extensively involved in the stress response to clubroot disease.

Phytohormones are crucial for plants to adapt in biotic and abiotic stresses through regulation of a wide range of adaptive responses. Studies reported that phytohormones play an important role in the pathogenesis of clubroot, especially in the formation of root tumors [17,18]. It was found that salicylic acid (SA) was the key plant hormone in the process of clubroot resistance. Exogenous SA could effectively inhibit the occurrence of clubroot disease in susceptible plants by inducing the production of pathogenesis-related proteins and regulated the activities of defense enzymes to make plants produce systemic acquired resistance [19,20,21]. In addition to SA, auxin, cytokinin (CK), and jasmonic acid (JA) also respond to the stress of clubroot. The contents of auxin, CK, and JA are increased significantly during the infection of roots of cruciferous plants. In *Arabidopsis thaliana*, the level of auxin increased due to the decrease in myrosinase at the early stage of disease development and the increasing expression of *Nitrilase* 1 and *Nitrilase* 2 genes in the later stage of disease development [22]. Up-regulation of *CK* genes and the down-regulation of CK *hydrogenas*/*deoxygenase* genes [23] led to a decrease in the CK content in *Arabidopsis thaliana* during the infection process. JA expression also is up-regulated during clubroot disease infection in *B. rapa* due to induction of the expression of *1-3-methylthiogluconic acid* (GSL) and *wax hydrolase*, which means that JA and auxin metabolic pathways may be related in the process of clubroot disease [14]. In resistant plants, several genes involved in the Ethylene (Et) pathway were down-regulated in the induction of the defense response [17,24]. In addition to the above phytohormones, abscisic acid (ABA) also plays an important role in plant disease resistance. ABA can coordinate plant resistance to drought stress and may promote plant disease resistance [25].

Both miRNAs and phytohormones are essential signaling molecules; they participate in most of the biological processes in the plant’s life cycle and a close relationship exists between them. miRNAs can regulate the content of plant hormones by regulating the genes related to phytohormones [26]. Although the functional pathways of hormones and miRNAs are different, they are sometimes involved in common biological process. For example, auxin precisely regulates the various processes of lateral root development through a *TIR1/AFB*-Aux/IAA-mediated transcriptional regulatory pattern [27], while miR167 also regulates lateral root development by regulating the crucial genes in the *ARF6/8* signal transduction pathway. Recently, some studies have emphasized the interaction between phytohormones and miRNAs, pointing that they can regulate each other in response to environmental stresses [18]. Some papers have listed the specific miRNAs that are regulated by several specific phytohormones [28,29].

Although there are some preliminary studies on the relationship between miRNAs and clubroot disease, most of them focus on the role of individual miRNAs; few studies have systematically analyzed the molecular mechanisms at the transcription level combined with plant hormones. Here, we explored the pathogenic mechanism by analyzing the high throughput sequencing of the transcriptome, miRNAs, degradome, and phytohormones of the resistant and susceptible materials at different days after inoculation to provide new insights for the disease resistant and sensing mechanisms of clubroot diseases at different stages.

## 2. Results

### 2.1. Phenotypic and Cytological Observation of R- and S-Lines

In the first 3 days after inoculation (DAI), there was no obvious symptoms of clubroot disease in susceptible and resistant lines. On 9 DAI, S-line material had clubroot symptoms but no such symptoms were present in the R-line. At 20 DAI, disease symptoms were fully developed in the S-line, which entered an advanced stage of disease. Roots of the S-line showed irregular swelling, while the roots of the R-line had no sign of swelling, suggesting that clubroot symptoms developed rapidly in the S-line over a period of 9–20 DAI (Figure 1a).

When assessing cross sections of paraffin-embedded roots, we did not observe any changes in the root cortex at 3 DAI; the roots of the S-line showed irregular enlargement and cell deformation at 9 DAI while the R-line did not show such changes; and at 20 DAI, the S-line showed obvious clubroot characteristics such as cell swelling and disorder of root cortical cells (Figure 1b).

### 2.2. Small RNA Sequencing Profile

To understand the mechanism and response of miRNA under *P. brassicae* infection, we preformed small RNA sequencing using the samples, which were collected in four different time points after inoculation. Sequencing data showed the reliability after filtration and statistical analysis (Appendix A). Statistics of filtered reads after evaluation and filtration for base quality and length are presented in Appendix A, which also conferred the reliability of the data. The length of miRNAs varied from 18 to 24 nt (Figure 2a). Most of the miRNAs recovered had 21 nt (43.6%), followed miRNAs with 24 nt (32.2%). S-line material had 2593 novel miRNAs, which covered 73% of the total miRNAs, while R-line material had 2340 novel miRNA and covered 76% of the total miRNAs (Appendix A). The number of known miRNAs of S- and R-lines was 968 and 731, respectively, which covered 27% and 24% of the total miRNAs (Figure 2b).

The miRNAs of each material were categorized into four groups based on four time points. Maximum miRNAs were expressed at 0 DAI in the S-line, and 256 miRNAs were expressed in all the four time points. While in the R-line, adequate miRNAs were expressed at 0 DAI and 174 miRNAs were expressed in all time points (Figure 2c).

### 2.3. Co-Analysis of the Differential Expression of the Transcriptome and miRNAs between R- and S-Lines under Different DAIs

To further analyze and compare differentially expressed miRNAs (DEmiRNAs) at different time points, high-throughput sequencing was used to obtain more statistics about miRNAs under the four time points (Appendix A). We also analyzed the differentially expressed genes (DEGs) at different time points (Appendix A). The number of DEmiRNAs of the S- and R-lines was varied at different time points under *P. brassicae* infection. But the number of DEmiRNAs in the S-line was changed more obviously than in the R-line. The quantity of DEmiRNAs in both materials changed over times and the difference between R- and S-lines was not obvious.

In the R-line, 9 DAI vs. 20 DAI had the most abundant miRNAs and all 13 DEmiRNAs were up-regulated. While 0 DAI vs. 3 DAI had the fewest DEmiRNAs with three up-regulated and three down-regulated. It was worth noting that in the analysis of DEGs, 0 DAI vs. 3 DAI had the largest quantity of DEGs, while 0 DAI vs. 9 DAI, 0 DAI vs. 20 DAI had abundant numbers of DEGs, which were contrary to result of DEmiRNAs (Figure 3a).

In the S-line, the analysis of DEmiRNAs was consistent with DEGs. The largest number of DEmiRNAs and DEGs were found in 9 DAI vs. 20 DAI. We found 27 up-regulated and 54 down-regulated of DEmiRNAs, whereas 2648 up-regulated and 4549 down-regulated of DEGs were observed. Furthermore, the distribution patterns showed differential regulation of miRNAs and genes in 3 DAI vs. 20 DAI., We identified 84 DEmiRNAs and 7683 DEGs at 9–20 DAI, but only 29 DEmiRNAs and 1874 DEGs at 3 DAI vs. 9 DAI (Figure 3b). Therefore, 9–20 DAI may be the critical time period that strongly influences miRNAs and gene expression under the infection of *P. brassicae*.

### 2.4. GO and KEGG Pathway Analyses of DEGs

GO and KEGG pathway analyses were conducted to understand the manifestation of the potential biological role of DEGs. We selected the time point of 0 DAI vs. 3 DAI in the R-line and 9 DAI vs. 20 DAI in the S-line as the critical time points for further analyses. From the GO enrichment, we found that the S- and R-lines had a similar distribution of biological process, cellular component, and molecular function categories. In the analysis of 0 DAI vs. 3 DAI in the R-line, the GO enrichment data showed most of the DEGs related to biological process were involved in “single-organism process”, “metabolic process”, and “cellular process”. In the cellular component part, the GO terms “cell part”, “cell”, and “organelle” accounted for a large proportion and rest of the DEGs were related to molecular function “binding”, “catalytic activity”, and “nucleic acid binding transcription factor activity” (Figure 4a). For KEGG annotations of pathway analyses; in the top 20 pathways, most of the DEGs were found enriched in “plant hormone signal transduction”, “plant-pathogen interaction”, and “phenylpropanoid biosynthesis”, of which “plant hormone signal transduction” was the richest. According to the enrichment data, “plant hormone signal transduction” and “plant-pathogen interaction” were the most critical enriched pathways (Figure 4c).

In the analysis of 9 DAI vs. 20 DAI in the S-line, the top three GO terms of biological process, cellular component, and molecular function were same as the result of 0 DAI vs. 3 DAI in the R-line (Figure 4b). For KEGG annotations of pathway analyses, most of the DEGs were found to be enriched in “phenylpropanoid biosynthesis” and “plant-pathogen interaction”. In the top 20 pathways, “phenylpropanoid biosynthesis” was the richest pathway. According to the enrichment data, “phenylpropanoid biosynthesis” and “plant-pathogen interaction” were found as the key pathways associated with the response to *P. brassica* (Figure 4d).

### 2.5. Phytohormone Analysis in R- and S-Lines at Four DAIs

From the previous studies, we know that cell division and cell enlargement were the main features of clubroot disease development, whereas plant hormones, like auxin and CK, accelerated cell division and cell enlargement. Therefore, we performed phytohormone quantification to figure out the relationship between phytohormones and clubroot development and to further understand the infection mechanism of clubroot.

Through targeted analysis of plant hormones, we mainly detected seven categories of plant hormones: JA, auxin, SA, ABA, CK, Et, and brassinosteroid (BR). Results showed that CKs included free cytokinin isopentenyladenine (iP), cis-zeatin (cZ), trans-zeatin (tZ), cis-zeatin (cZ), cis-zeatin riboside (cZR), and bound cytokinin isopentenyladenosine (iPA). JAs included jasmonic (JA), jasmonoy l-L-isoleucine (JA-Ile), and cis-12-oxo-phytodieneic acid (cis-OPDA). Brassinolides (BRs) included typhasterol (TY). Ets included synthetic precursors of aminocylopropane-1-carboxylic acid (ACC). We estimated higher phytohormone content in the S-line than the R-line. In the R-line, most phytohormones started to decline from 3 DAI, while in the S-line, they started to decline at 9 DAI indicating that 3 DAI and 9 DAI were the crucial time points for resistance and susceptibility mechanisms regarding phytohormones production.

In the R-line, the changes in the content of auxins, Ets, CKs, and BRs showed obvious similarity in the treated group. The expression of these plant hormones was increased at 0–3 DAI and decreased at 3–20 DAI. SA and JA reached the maximum level at 0–3 DAI; the hormone level started to decrease at 3–9 DAI and increased again at 9–20 DAI. ABA content was decreased at 0–20 DAI. All hormone levels, except ABA, peaked at 3 DAI, which indicated that the response of plant hormones to clubroot is inconsistent but the critical time of expressing resistance was at 3 DAI (Figure 5). In the S-line, no difference was found in the hormone levels of BRs and SA between the treated and control groups. The level of auxin in the treated group was increased at 9 to 20 DAI, while in the control group, it was decreased at this time point. The CKs showed a clear upward trend at 3–9 DAI, but the upward speed was decreased at 9–20 DAI (Figure 5). The level of Ets also showed a downward trend, while JA showed an upward trend at 9–20 DAI. This suggests that the increase in auxin may accelerate the onset speed by altering the change of other hormones. According to the above results, it can be concluded that the time points of altering different hormones to the disease is different, but 9–20 DAI may be a key period of hormone synergy.

### 2.6. miRNAs and Target Genes of the S-Line under P. brassicae Infection

To understand the infection mechanism, we considered the sequence data of 9–20 DAI as the major time points for detailed analyses. Transcriptomics data at 9 DAI vs. 20 DAI showed 82 DEmiRNAs and 1318 differentially expressed target genes (Appendix A). After co-analysis of the transcriptome and degradome, 42 miRNAs with 13 targets genes (Appendix A) were found as paired. A heatmap was drawn by using miRNAs and their target genes and found that some DEmiRNAs and their target genes showed a similar expression pattern (Appendix A). Among them, six highly expressed DEmiRNAs (*bra-miR319-3p*, *bra-miR164b-5p*, *bra-miR164e-5p*, *novel_miR142*, *novel_miR91*, and *bra-miR167a*) and six DEmiRNAs and their associated target genes (*Bra004125*, *Bra023671*, *Bra020236*, *Bra021926*, *Bra028685*, and *Bra029142*) were selected to perform real-time fluorescent quantitative PCR (qRT-PCR) to verify the sequencing results (Appendix A). The expression analysis of the genes by qRT-PCR was perfectly matched with the sequence data, which confirmed the reliability of the sequence data. We found that four miRNAs had a positive relationship with their targets, and two paired miRNA–targets were negatively related to their targets. Four miRNA-target pairs; *bra-miR164b-5p/Bra028685*, *bra-miR164e-5p/Bra028685*, *novel_miR_142/Bra020236*, and *novel_miR_91/Bra020236* showed a similar expression tendency. All of them were highly expressed at 0–3 DAI, gradually decreased at 3–9 DAI, and then increased at 9–20 DAI. While the *bra-miRNA167a* was highly expressed at 0 DAI, decreased quickly at 3–9 DAI, and then slowly increased at 9–20 DAI. Target genes *Bra004125* showed high expression at 9 DAI; an increasing tendency was found at 0–9 DAI and a decreasing tendency at 9–20 DAI. The *bra-miR319a-3p* and its target *Bra018280* gene also showed negative regulation (Figure 6). Based on the above result, we hypothesized that these highly expressed DEmiRNAs may paly crucial roles in the plant’s infection response mechanism to *P. brassicae*. To figure out the function of miRNA–target pairs under infection with *P. brassicae* at 9–20 DAI in the S-line, a co-expression network of miRNAs, target genes, gene families, and plant hormones was established in the cytoscape platform. The co-expression network included 26 DEmiRNAs, eight target genes of DEmiRNAs, and four phytohormones responding to *P. brassicae*. The network showed that the eight stress-responsive genes encoded the auxin response factor (*ARF*), *Arabidopsis thaliana* homeobox protein (*AtHB*), total circulating protein (*TCP*), nascent polypeptide-associated complex (*NAC*), squamosa promoter binding protein-like (*SPL*), and revoluta (*REV*) (Figure 7).

## 3. Discussion

Figuring out the infection mechanism of *P. brassicae* could help in taking more effective control measures against the clubroot disease of cruciferous crops. We combined the miRNAs, degradome, transcriptome, and plant hormones to figure out the critical periods of resistance and susceptibility during the infection mechanism.

In this study, we reported that the critical time points of the resistance and susceptibility reactions were 0–3 DAI and 9–20 DAI, respectively. To further elucidate the infection mechanism, we selected the DEGs and DEmiRNAs at 9- 20 DAI in S-line material and compared them with the degradome for identifying the key genes and miRNAs that played a key role in determining critical infection period. Thirteen key genes were selected, and among them, eight genes (*Bra001480* (*NAC1*), *Bra004125* (*ARF8*), *Bra010949* (*SPL10*), *Bra020236* (*REV*), *Bra021926* (*AtHB*), *Bra028685* (*NAC4*), *Bra030820* (*NAC1*), and *Bra018280* (*TCP10*)) are related to phytohormone regulation. We also selected 26 miRNAs that are associated with the above genes. Among them, *bra-miR319a-5p*, *bra-miR167*a, and *bra-miR164a/b/e-5p* were the known miRNAs that regulate plant hormone-related genes during pathogen infection.

*Bra021926* (*AtHB*) and *Bra020236* (*REV*) were targeted by the same novel miRNAs, which are identified in this study. *AtHB* and *REV* are related to Et, auxin, and CK regulation in plants [18]. But the specific regulatory mechanism is not clear. *Bra001480* (*NAC1*), *Bra028685* (*NAC4*), and *Bra030820* (*NAC1*) are targeted by three miRNAs, *bra-miR164e-5p, bra-miR164c-5p,* and *bra-miR164b-5p*. These three miRNAs belong to the *miR164* family. Previous studies have been shown that the miRNA–target pair *miR164*–*NAC* was relevant to auxin and Et, and both are the most critical plant hormones in the process of infection by *P. brassica. NAC1* is part of the NAC-domain gene family and the first *NAC* gene reported to be involved in root development. *NAC1* is specifically involved in the control of lateral root initiation. *NAC1* mediates auxin signaling to promote lateral root development and control lateral root initiation [30,31]. Overexpression of *NAC1* can promote the lateral root development of plant, while exogenous auxin application reduced the number of lateral roots [32] indicating that high expression of *NAC1* can inhibit auxin synthesis. *miR164* functions as a negative regulator of auxin-mediated lateral root development by controlling *NAC1* at the miRNA level [33]. *miR164* also negatively regulates the Et synthesis precursor ACC. Et can mediate root elongation by influencing the biogenesis and distribution of auxin in roots. [34]. The miRNA–target pair *miR164*–*NACs* may be critical regulators upon infection of *P. brassica*. We found that *Bra018280* (*TCP10*) was targeted by *bra-miR319a-5p*. A previous study reported that the miRNA–target pair *miR319a*–*TCP10* was relevant to auxin, Et, JA, and CK, which are the most critical plant hormones in the process of infection by *P. brassica* [35]. The target gene of *bra-miR319* is *TCP*, which encodes plant-specific transcription factors [36]. As a non-coding miRNA, *bra-miR319a* does not function by itself but acts by targeting downstream *TCP* transcription factors to influence the auxin signaling and inhibit the synthesis of IAA [36,37]. In our analysis, we found *Bra004125* (*ARF8*) was targeted by *bra-miR167a*. In *Arabidopsis*, *bra-miR167* and *ARF8* are expressed in the pericycle of roots, where they can mediate a pericycle-specific response to nitrate treatment and subsequent development of lateral roots [38]. In addition, *miR167*a may control nodule formation and root architecture by suppressing auxin signaling, supporting the notion that attenuated auxin signaling favors nodulation in legume plants [39,40].

This study confirmed that the formation of swollen roots after being infected by *P. brassicae* mainly depends on two hormones: auxin and CK. The formation of swollen roots depends on the proportion of the two hormones in the progress of disease. The high proportion of auxin/CK can contribute formation of gall [14]. At the early stage of infection, the levels of CK increased while the levels of CK decreased at the later stages [41]. Auxins are well-known phytohormones for their pivotal role in plant development and defense responses [42]. Auxins can influence cell division, enlargement, differentiation, and polarity to contribute to gall initiation and gall expansion. Et has been shown to strongly inhibit the elongation of the root in plants [43]. Et and auxin can regulate each other; Et can regulate the key genes involved in auxin synthesis to maintain auxin levels, while auxin also upregulates ACC synthase to modulate Et levels in plants [44]. Both auxin and Et can inhibit root elongation and lateral root development, which can contribute to the formation of gall in the root [45]. Previous studies reported that JA expression is up-regulated after *P. brassicae* infection because JA induces the expression of 1-3-methylthiogluconic acid (GSL) and wax hydrolase, which is the key material for auxin synthesis [46]. It is possible that the JA metabolic pathway and auxin metabolic pathway may be related during the occurrence of clubroot disease. The trend of changes of their content in the R- and S-lines in this study are consistent with previous studies.

Combining these results with previous reports, we proposed a model of the disease susceptibility mechanism regulated by miRNAs and their target genes and plant hormones upon infection of *P. brassicae* at the late stage. To draw the schematic diagram, we used the identified key miRNAs *bra-miR164*, *bra-miR319,* and *bra-miR167,* as well as their target genes related to phytohormone regulation. During infection, *bra-miR319* and *bra-miR167* were inhibited, and their targeted genes, *TCP10* and *ARF8*, were highly expressed, while both *bra-miR164* with its targeted genes, *NAC1* and *NAC4,* were highly expressed. Highly expressed genes, *TCP10*, *ARF8,* and *NAC1/4,* contributed to IAA synthesis. The highly expressed auxin promoted the ACCs Et and JA, but inhibited the CKs tZ, czR, and tzR. Increases in auxin can contribute to cell division, gall initiation, and gall expansion. The Et can inhibit the root elongation and lateral root growth and JA can increase the susceptibility of the plant. All these together can promote the formation of swollen roots. In addition, the low expression of the CKs tZ, czR, and tzR in the late infection phase also can inhibit the cell elongation to promote the formation of swollen roots (Figure 8). Overall, the schematic diagram of infection regulation mechanism shows the relationship of miRNAs with targeted genes, miRNAs with plant hormones, and plant hormones with plant hormones. Three kinds of relationships indicate that the infection of *P. brassicae* in crucifer plants is complex. Upon analysis of the late infection phase, we found that the miRNA–targets *bra-miR164*–*NAC1/4*, *bra-miR319*–*TCP10,* and *bra-miR167*–*ARF8* were closely related to the plant hormones, which are associated with clubroot symptom development. Nevertheless, although most of the plant hormones associated with *P. brassicae* were studied thoroughly, the relationship between the plant hormones remains unclear.

In this study, we did not study each miRNA individually, we only assessed out the specific role of miRNAs in the regulatory network of plant hormones. Research shows that miRNAs play an important role in other metabolic pathways, such as PTI and ETI. Therefore, more investigations should be performed to figure out the relationship between the miRNAs and metabolic pathways, and more studies should be carried out to explore the role of miRNAs in the plant immune system as well as clubroot development in Chinese cabbage.

## 4. Materials and Methods

### 4.1. Plant Materials and P. brassicae Treatment

Two contrasting *B. rapa* materials, the BrT24 (R-line) and Y510-9 (S-line), regarding *P. brassicace* infection were collected from the Institute of Horticulture, Henan Academy of Agricultural Sciences, China. The *P. brassicae* strain used in this study was obtained from the clubroot of *B. rapa* in Xinye County, Henan Province, China (113.97° E, 35.05° N), and was identified as race 4 by the Williams system [47]. The *P. brassicae* spore suspension was diluted to 1 × 10^7^/mL and was used as inoculum to infect plants with resistant and susceptible materials. Seeds of BrT24 and Y510-9 were planted in a 50-hole acupoint tray and placed in the incubator with 25/20 °C and 16 h/8 h (light/dark). After 20 days, each hole was inoculated with 20 mL of *P. brassicae* solution. Root samples were collected at four different time points, before inoculation (0 days after inoculation, 0 DAI), at the time of cortex infection (3 DAI), early onset (9 DAI), and the later stage (20 DAI) of the disease development. The root of the R-line and S-line was cut at different inoculation stages for paraffin sectioning [48], and the Leica DM48 microscope (Leica, Weztlar, Germany) was used to observe the infection of root cells.

### 4.2. RNA Extraction and Construction of the cDNA Library

RNA was extracted from the root samples of S- and R-line materials, which were collected at 0 DAI, 3 DAI, 9 DAI, and 20 DAI following the Trizol method [14,49]. First, Nanodrop, Qubit 2.0, and Aglient 2100 methods were used to detect the purity, concentration, and integrity of RNA samples to ensure sample quality. After the sample was qualified, the total RNA was used as the starting sample, and the small RNA Sample Pre Kit (TIANGEN, Beijing, China) was used to construct the library. T4 RNA Ligase 1 and T4 RNA Ligase 2 (truncated) were used to connect adapters at the 5′ end and 3′ end of the small RNAs, respectively, which were reverse transcribed and cDNA synthesized for PCR amplification. Polyacrylamide gel electrophoresis (PAGE) was used to screen the target fragment; the gel was cut to recover the fragments as the small RNA library. After the library was constructed, the concentration and insert size of the library were detected using Qubit2.0 and Agilent 2100, respectively, and the effective concentration of the library was accurately quantified using the qPCR method to ensure the quality of the library. After passing the library inspection, Illunima HiSeq X-ten was used for high-throughput sequencing. The raw image data files obtained after sequencing were transformed into raw sequencing through base recognition analysis, namely raw reads. To ensure the quality of information analysis, raw reads were filtered out and low-quality reads were removed to obtain clean reads, which were then used for subsequent analyses.

### 4.3. Transcriptome Sequencing and Assembly Analysis

The cDNA library was constructed from the pooled RNA samples of S- and R-lines, which was sequenced by using the Illumina HiSeq2500 platform according to Illumina paired-end RNA-seq. The low-quality reads were trimmed out and overall clean reads were acquired for analysis. The number of mapped reads in the samples and the length of the transcript were estimated. The fragments per kilobase of transcript per million (FPKM) [50] was used as an indicator for measuring the level of gene expression, and DESeq2 was used for differential expression analysis among different samples. In the analysis of DEGs, the fold change (FC) was used to indicate the rate of expression between two samples. The false discovery rate (FDR) was estimated by using the correct *p*-value (*p*-Value = 0.05). Genes with FC ≥ 2 and FDR < 0.01 were used as DEGs.

### 4.4. Identification of Known and Novel miRNAs and Predication of Their Target Genes

Clean reads were compared to the Chinese cabbage reference genome and the mature sequence of miRNAs of the miRBase database (version 22.0), the identified reads were regarded as known miRNAs. The novel miRNAs were predicted by using the classic Bayesian score considering the known unattended miRNA sequence, the starting point of the miRNA transcription and its preliminary sequence, iconic hairpin structure, other biological characteristics, and matching the possible precursor sequence with miRDeep2 [51]. Target genes associated with known miRNAs and novel miRNAs were finalized by using the software Target Finder [52].

### 4.5. Degradome Sequencing

The total RNA extracted from the roots of S- and R-line materials was equally mixed separately, and pooled samples were used to construct the degradation group library. The construction and sequencing of the library was completed by Bamako Biotechnology Co., Ltd. Beijing, China. First, mRNAs were captured from total RNAs by magnetic beads, then a 5′ linker was ligated and reverse transcribed to cDNA by using random primers. After PCR amplification, the target fragment was purified by PAGE and the constructed library was sequenced with Illumina HiseqTM 2500. The original tags from each sample were sequenced to remove linker sequences and low-quality sequences were filtered out to obtain clean and cluster tags. Cluster tags were compared to the reference genome of Chinese cabbage to get the distribution of tags on the genome. Cluster tags were compared to the RFAM database [53] and non-coding RNAs were identified. The unannotated sequence was used for analyzing degradation site. After comparing cluster tags with the Rfam database, non-coding RNAs were annotated while the rest were used for subsequent degradation position analysis. Information from the miRNA database and the sequence of predicted miRNAs was used to detect the degradation site by Clearland3 software, with the condition *p* < 0.05. The target genes were compared with NR, Swiss-Prot, GO, Kegg, and COG databases by BLAST software to obtain the annotation information of target genes.

### 4.6. Quantification of Phytohormones

Samples were taken 20, 23, 29, and 40 days after planting (0, 3, 9, and 20 DAI). The root samples were collected from plants of S- and R-line materials, which were inoculated and not inoculated at the corresponding time. Eight milligrams of the sample and 50 mL of internal standard solution were added in a 2 mL centrifuge tube. Thereafter, the samples were grinded in liquid nitrogen. One milliliter of acetonitrile aqueous solution (1% FA) was added and shaken for 2 min to ensure it is mixed well and was kept for 12 h at 4 °C in the dark for extraction of phytohormones. After that, the sample was centrifuged for 10 min at 14,000× *g*, the supernatant was dried with 800 μL and was added with 200 mL acetonitrile water (1:1, *V*/*V*). The sample was further centrifuged for 10 min at 14,000× *g* and the supernatant was taken for analysis. Phytohormones were separated using Waters I-Class LC high-performance liquid chromatography system (Waters, Milford, United States) and was analyzed by using a 5500 QTRAP mass spectrometer (AB SCIEX, Shanghai). The ion pairs were detected by the selective reaction/multiple reaction monitoring (MRM) technique using Thermo Scientific Xcalibur 2.1 (Thermo Fisher Scientific, San Jose, CA, USA). Finally, Multiquant software was used to extract the peak area and retention time. According to the standard curve, the content of plant hormone was calculated.

### 4.7. qRT-PCR Validation of miRNAs and mRNAs

To verify the reliability of the high-throughput sequencing data, miRNAs and mRNAs were chosen randomly to perform qRT-PCR. In this study, qRT-PCR was used to detect miRNAs by tail-adding method, and the selected miRNAs were detected by qRT-PCR using Takara’s kit (Mir-X™ miRNA First-Strand Synthesis and TB Green™). mRNA reverse transcription was achieved by using Takara’s reverse transcription kit (PrimeScript™RT reagent Kit with gDNA Eraser), and qPCR with cDNA was performed using Takara’s real-time Quantitative PCR kit. The primers were designed on the prime 5.0 software platform and synthesized by SunYa Biological Company (SunYa Biological Company, Zhengzhou, China).

## 5. Conclusions

In this study, we analyzed the data of the transcriptome, small RNAs, degradome, and phytohormones of R- and S-line materials of Chinese cabbage to figure out the infection regulation mechanism under the influence of plant hormones upon *P. brassicae* infection. We found that the critical time point of resistance and infection was different. The key time of resistance was at 0–3 DAI, while time of infection was 9–20 DAI. The time of response of plant hormones to *P. brassicae* also differed; the crucial plant hormone response time point of resistance was 3 DAI, while that of infection was 9 DAI. It is worth to noting that in the S-line material, the crucial time of the plant hormone response may be different, with a trend in some plant hormones changing at 3 DAI. The result indicates that the plant hormone response mechanism of infection in the S-line material may more complex than resistance. Through multiple comparisons of data of crucial time point at 9–20 DAI in the S-line, we screened out eight key genes, *Bra001480 (NAC1)*, *Bra004125 (ARF8)*, *Bra010949 (SPL10)*, *Bra020236 (REV)*, *Bra021926 (PHB)*, *Bra028685 (NAC4)*, *Bra030820 (NAC1)*, and *Bra018280 (TCP10),* and five critical miRNAs *bra-miR164e-5p*, *bra-miR164c-5p*, *bra-miR164b-5p*, *bra-miR167a,* and *bra-miR319a,* that may be worth further study. In a word, our study could deepened the understanding of the infection regulation mechanism of *P. brassicae.*

## Figures and Tables

**Figure 1 ijms-24-02414-f001:**
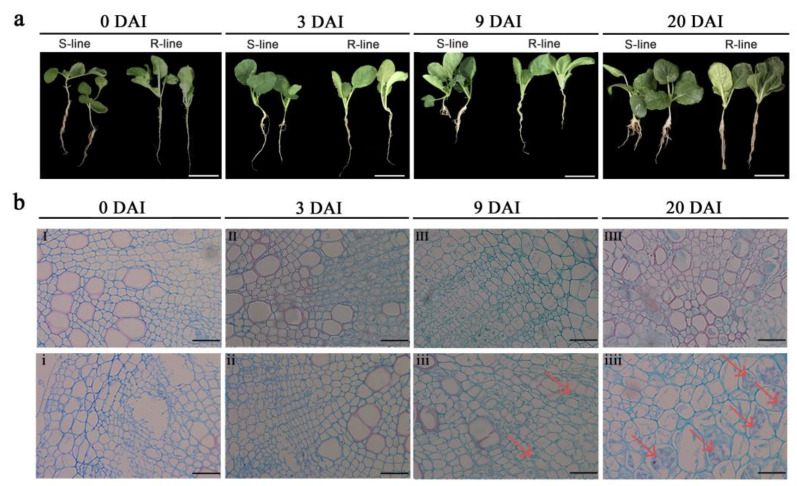
(**a**) Representative images of S- and R-line material at 0, 3, 9, and 20 DAI (days after inoculation) of *P. brassicae.* Bar = 5 cm. (**b**) Cross sections (4 µm thick) of paraffin embedded roots of the R-line (I–IIII) and S-line (i–iiii) at 0, 3, 9, and 20 DAI, respectively. Bar = 50 μm. The red arrow points to abnormal cells after clubroot infection.

**Figure 2 ijms-24-02414-f002:**
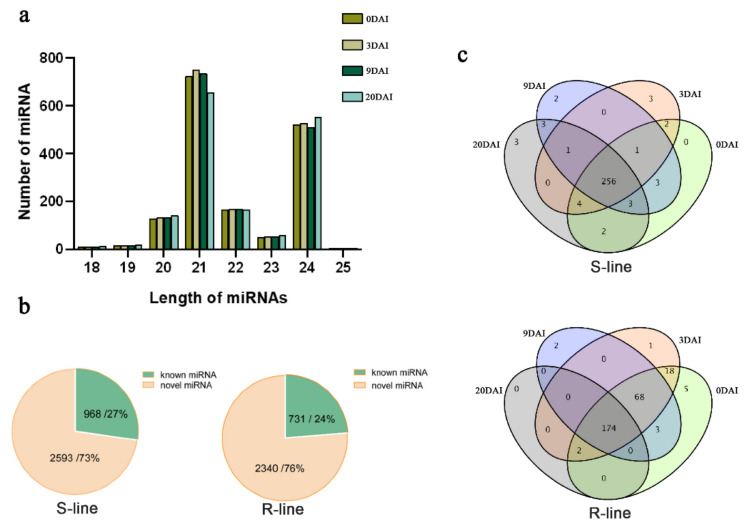
(**a**) Number of miRNAs distributed by different lengths found in different development periods of the two materials at 0, 3, 9, and 20 DAI. (**b**) The distribution of known and novel miRNAs in the R-line and in S-line. (**c**) Venn diagrams of differentially expressed miRNAs in different development periods in the two materials at 0, 3, 9, and 20 DAI.

**Figure 3 ijms-24-02414-f003:**
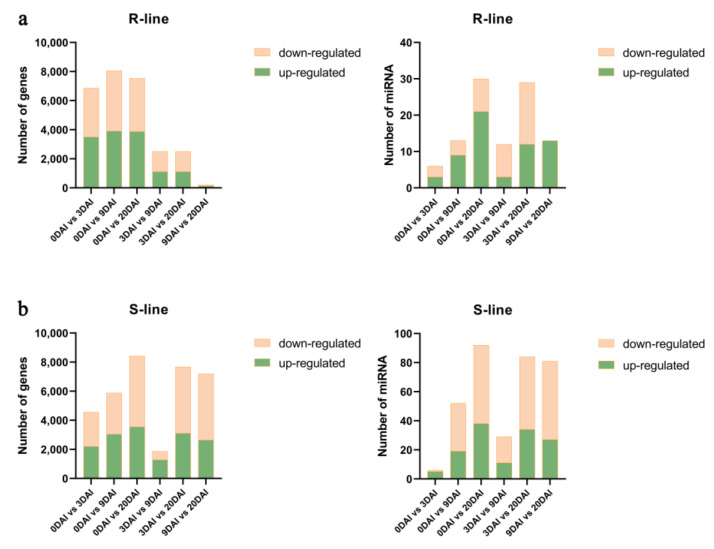
(**a**) Number of up- and down-regulated DEmiRNAs and DEGs in the R-line material BrT24 and (**b**) number of up- and down-regulated DEmiRNAs and DEGs in the S-line Y510-9 in different development periods.

**Figure 4 ijms-24-02414-f004:**
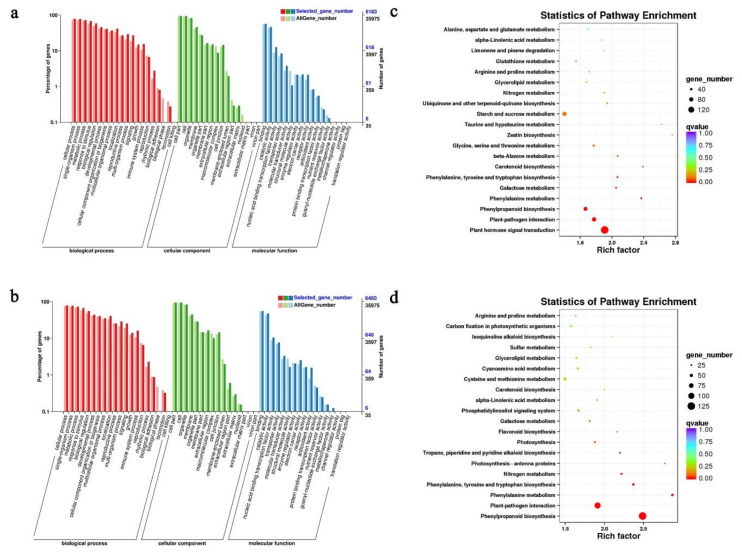
(**a**) GO enrichment of DEGs of the R-Line and (**b**) S-line, and (**c**) KEGG pathway enrichment of DEGs of the R-line and (**d**) S-line.

**Figure 5 ijms-24-02414-f005:**
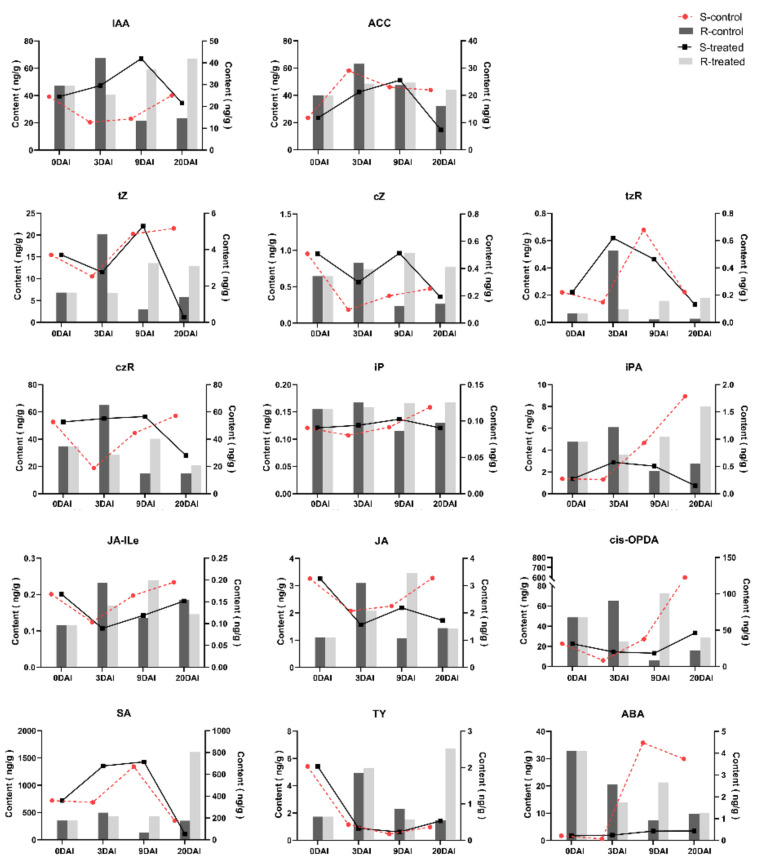
The contents of different phytohormones in the roots of R- and S-lines at four time points after inoculation with *P. brassicae* at 0, 3, 9, and 20 DAI. The content of phytohormones in the S- and the R-lines are presented in the left and right Y axis, respectively.

**Figure 6 ijms-24-02414-f006:**
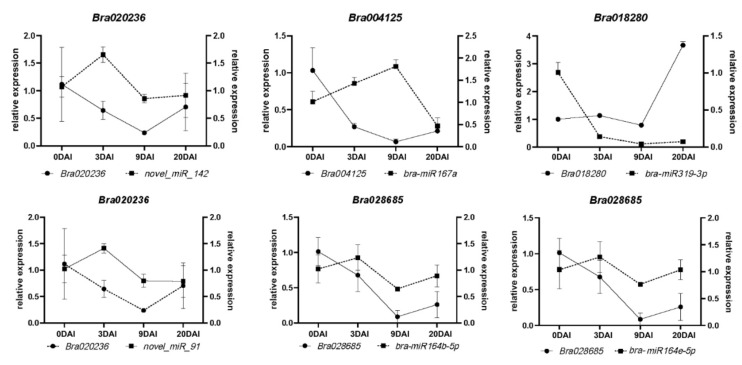
qRT-PCR validation of miRNAs and their targeted gene expression at different development periods of the S-line after inoculation. The dotted and solid lines are expressed miRNAs and targeted genes, respectively.

**Figure 7 ijms-24-02414-f007:**
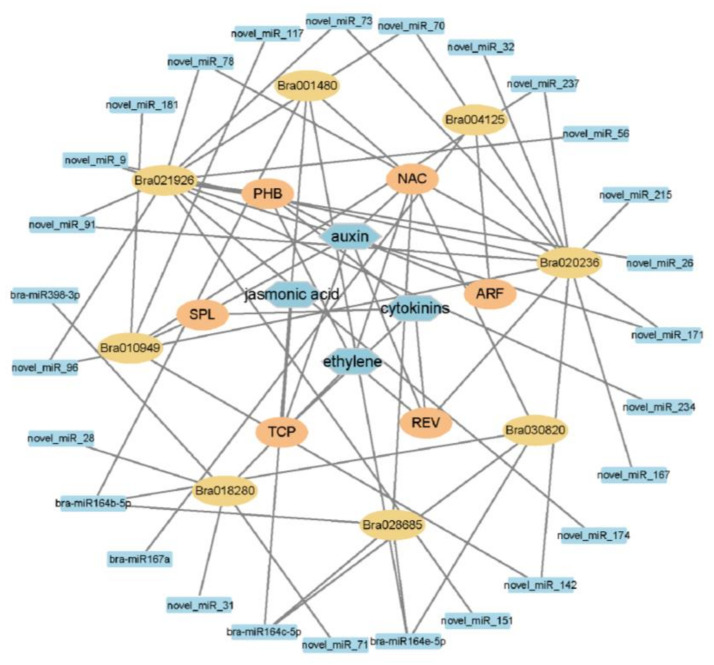
The miRNA–mRNA co-expression network for genes related to phytohormones.

**Figure 8 ijms-24-02414-f008:**
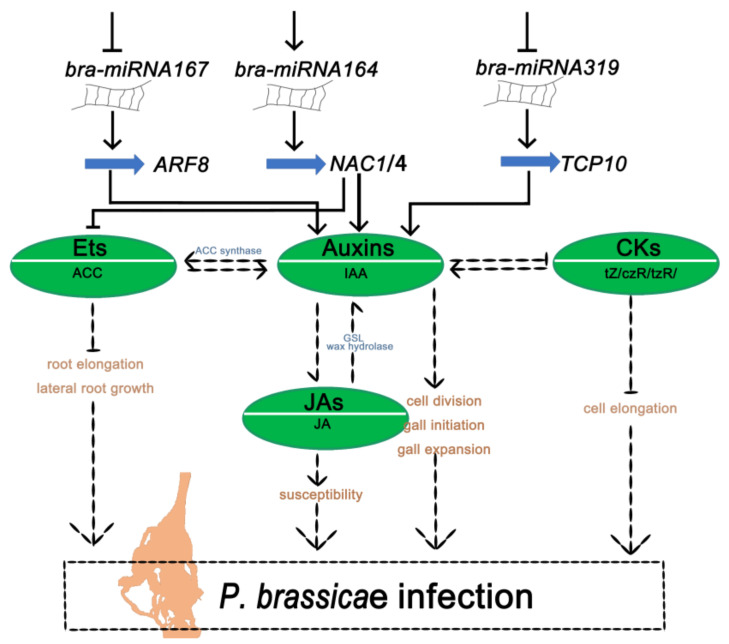
Proposed model of infection response to *P. brassicae* in *B. rapa* of the S-line based on hormonal regulation after pathogen inoculation.

## Data Availability

The original contributions presented in the study are publicly available. This data can be found here: National Center for Biotechnology Information (NCBI) BioProject database under accession number PRJNA868821 and PRJNA743585.

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
