# Peer review of "Integrative Transcriptome, miRNAs, Degradome, and Phytohormone Analysis of Brassica rapa L. in Response to Plasmodiophora brassicae"

_ijms, 2023, doi:10.3390/ijms24032414_

Round 1
Reviewer 1 Report
Line 30: add to keywords "Brassica rapa"
Line 254: thaliana
Line 366: brassicae
All scientific names in references need revision to write italics.
Author Response
Response to Reviewer 1 Comments
Dear Editor,
I would like to extend my gratitude to the respected reviewers, who spend your valuable time to review our manuscript. Your valuable comments were necessarily important for substantial improvement of our manuscript. The given comments and suggestions which we try to address point-by- point as follows:
Point 1: Line 30: add to keywords "Brassica rapa"
Response 1: Thanks for your comments. We have already added it to the keywords.
Point 2: Line 254: thaliana
Response 2: We have already corrected that.
Point 3: Line 366: brassicae
Response 3: We have corrected that.
Point 4: All scientific names in references need revision to write italics.
Response 4: We are sorry for the mistake and thanks for your careful and rigorous, we have corrected that.

Reviewer 2 Report
In this MS, the author performed comprehensive analysis of integrative transcriptome, small RNA, degradome and phytohormone to explore the infection mechanism of P. brassicae, and proposed a model of disease susceptibility mechanism regulated by miRNAs and their target genes and plant hormones upon infection of P. brassicae at the late stage. It is important to the scientific world in the field of infection mechanism of P. brassicae. However, there are many issues where the author can concentrate while revising the manuscript prior to its final acceptance.
Introduction
What is the research actuality of miRNA in infection/resistance mechanisms of clubroot disease?I think this is an important part of introduction.
Results
Line 50 Arabidopsis should in italic
Line 110 the author performed small RNA sequencing,is there any difference in other kinds of sRNA? If the author only analysis miRNA, I think it is better to use “miRNA” in this sentence.
Line 117 what dose “new miRNAs” mean here,never reported before or not detected in another line?
In Figure 3 “(a)” was missing
Line 141-155 is there a list of the up- and down-regulated DEmiRNAs and DEGs?
Line 163 “Form” or “From”?
Figure 4 (a) and (b) are not clear enough.
Figure 5 what does the left and right Y axis indicated, respectively? Does it indicate two lines? All the figures with two Y axis should describe which index each axis stand for.
What dose Y-control, B-control, Y-treated, B-treated mean?
Page 9 supplementary figure cannot be shown in main text.
Discussion
The author should discuss the improvement of your findings to the infection mechanisms of clubroot disease, and the relationship between your findings and previous studies.
Line 359-362 What dose it mean?
Author Response
Response to Reviewer 2 Comments
Dear Editor,
I would like to extend my gratitude to the respected reviewers, who spend your valuable time to review our manuscript. Your valuable comments were necessarily important for substantial improvement of our manuscript. The given comments and suggestions which we try to address point-by- point as follows:
Point 1: Extensive editing of English language and style required
Response 1: Thanks for your comment. We are extremely sorry for your inconvenient to read the manuscript due to linguistic error. The manuscript has been revised extensively for English language by a native English speaker and resolved most of the error.
Point 2: What is the research actuality of miRNA in infection/resistance mechanisms of clubroot disease?I think this is an important part of introduction.
Response 2: Thanks for your comments. We have already added the part inside the introduction. See in line 50-60.
Point 3: Line 50: Arabidopsis should in italic
Response 3: We have already corrected that.
Point 4: Line 110: the author performed small RNA sequencing,is there any difference in other kinds of sRNA? If the author only analysis miRNA, I think it is better to use “miRNA” in this sentence.
Response 4: We have corrected that.
Point 5: Line 117:what dose “new miRNAs” mean here,never reported before or not detected in another line?
Response 5: We are sorry for the mistake and thanks for your careful and rigorous, we have corrected the wrong statement and substituted “new miRNA” for “novel miRNA".
Point 6: In Figure 3 “(a)” was missing
Response 6: We have corrected that.
Point 7: Line 141-155: is there a list of the up- and down-regulated DEmiRNAs and DEGs?
Response 7: Thanks for your comments, we have already added this content in supplementary materials.
Point 8: Line 163: “Form” or “From”?
Response 8: We are sorry for the mistake and thanks for your careful and rigorous, we have corrected.
Point 9: Figure 4 (a) and (b) are not clear enough.
Response 9: We have corrected.
Point 10: Figure 5 what does the left and right Y axis indicated, respectively? Does it indicate two lines? All the figures with two Y axis should describe which index each axis stand for.
Response 10: Thanks for your comments. We have added the explanation of “two Y axis” to the illustration of Figure 5.
Point 11: What dose Y-control, B-control, Y-treated, B-treated mean?
Response 11: Thanks for your comments, we've changed this expression to be consistent with the previous text and replaced “Y-control, B-control, Y-treated, B-treated" with “S-control, R-control, S-treated, R-treated”.
Point 12: Page 9 supplementary figure cannot be shown in main text.
Response 12: Thanks for your comments, we have deleted the supplementary figure in main text.
Point 13: The author should discuss the improvement of your findings to the infection mechanisms of clubroot disease, and the relationship between your findings and previous studies.
Response 13: Thanks for your comments. We have added some sentence to the discussion: “In this study, we no longer study each miRNA individually but to find out the specific role of miRNAs in the regulatory network of plant hormone. Nevertheless, most of the plant hormones associated to P. brassicae were been studied thoroughly but the relationship between the plant hormones still not clear. And research shows that miRNA also plays an important role in other metabolic pathways, PTI and ETI“.
Point 14: Line 359-362 What dose it mean?
Response 14: We are sorry for the mistake and thanks for your careful and rigorous, we have deleted the sentence.

Round 2
Reviewer 1 Report
Accept in present form. Many thanks for your efforts.
Reviewer 2 Report
The MS have been carefully modified and improved,and could be accepted in present form.